# Characterizing the Core-Shell Architecture of Block Copolymer Nanoparticles with Electron Microscopy: A Multi-Technique Approach

**DOI:** 10.3390/polym12081656

**Published:** 2020-07-25

**Authors:** Vitalii Tkachenko, Loïc Vidal, Ludovic Josien, Marc Schmutz, Julien Poly, Abraham Chemtob

**Affiliations:** 1Université de Haute-Alsace, CNRS, IS2M UMR7361, F-68100 Mulhouse, France; vitalii.tkachenko@uha.fr (V.T.); loic.vidal@uha.fr (L.V.); ludovic.josien@uha.fr (L.J.); julien.poly@uha.fr (J.P.); 2Université de Strasbourg, CNRS, ICS UPR22, F-67034 Strasbourg, France; marc.schmutz@ics-cnrs.unistra.fr

**Keywords:** core-shell, TEM, cryo-TEM, SEM, positive staining, negative staining, PISA, nanoparticles

## Abstract

Electron microscopy has proved to be a major tool to study the structure of self-assembled amphiphilic block copolymer particles. These specimens, like supramolecular biological structures, are problematic for electron microscopy because of their poor capacity to scatter electrons and their susceptibility to radiation damage and dehydration. Sub-50 nm core-shell spherical particles made up of poly(hydroxyethyl acrylate)–*b*–poly(styrene) are prepared via polymerization-induced self-assembly (PISA). For their morphological characterization, we discuss the advantages, limitations, and artefacts of TEM with or without staining, cryo-TEM, and SEM. A number of technical points are addressed such as precisely shaping of particle boundaries, resolving the particle shell, differentiating particle core and shell, and the effect of sample drying and staining. TEM without staining and cryo-TEM largely evaluate the core diameter. Negative staining TEM is more efficient than positive staining TEM to preserve native structure and to visualize the entire particle volume. However, no technique allows for a satisfactory imaging of both core and shell regions. The presence of long protruding chains is manifested by patched structure in cryo-TEM and a significant edge effect in SEM. This manuscript provides a basis for polymer chemists to develop their own specimen preparations and to tackle the interpretation of challenging systems.

## 1. Introduction

One major interest of amphiphilic block copolymers lies in their ability to form aggregates of high morphological complexity. In the literature, there is an abundance of studies using amphiphilic block copolymers of different compositions and block lengths that are able to self-assemble into a variety of supramolecular aggregates, such as spheres, rods, lamellae, and vesicles [1]. The manipulation of interfacial curvature plays a central role in the control of aggregate morphology. It should be noted that not all these block copolymer supramolecular structures can be deemed nano-objects (or nanoparticles)—only those whose three dimensions span from approximately 1 to 100 nm can be [2]. Spherical nanoparticles are the most common and investigated copolymer nano-objects [3]. Among the plethora of methods developed for the formation of spherical, amphiphilic block copolymer nanoparticles, nanoprecipitation and polymerization-induced self-assembly (PISA) are the most popular. Nanoprecipitation involves the dissolution of a preformed copolymer in an organic water-miscible solvent. Upon addition to the aqueous phase, the organic solvent immediately diffuses out, thus leading to the formation of nanoparticles. In PISA, a lyophilic block is chain-extended using a monomer whose corresponding homopolymer is insoluble in the chosen solvent [4]. After reaching some critical degree of polymerization (DP), the second block eventually becomes insoluble, thus resulting in a spontaneous copolymer self-assembly and the formation of block copolymer nanoparticles.

Regardless of the synthetic pathway, block copolymer nanoparticles have attracted much attention for a diverse range of applications [5] including therapeutics delivery [6,7], imaging contrast agents [8], nanostructured films [9], stimuli-responsive nanomaterials [10,11], and the templating of inorganic materials [12,13]. Crucial for many applications is the core-shell architecture in which the inner region—mainly composed of the lyophobic blocks of the amphiphile—is surrounded by a brush-like corona rich in lyophilic blocks [14]. Various examples have shown a strong link between core-shell morphology and nanoparticles’ properties. In the field of nano-delivery systems, for instance, the inner core is thus able to act as cargo space for the solubilization of lipophilic drugs, proteins, or nucleic acids [15]. In bioimaging, the hydrophobic core can serve as nano-depot for the accommodation of a fluorescent agent [16]. The engineering of the shell can be also utilized to attach bioactive ligands (antibodies, peptides, etc.) in order to facilitate cell entry for targeted drug delivery but also to prepare supported catalyst [17] or diagnostic systems [18]. Finally, latexes containing a soft shell and a hard core can be used as building block for the creation of nanostructured coatings with improved mechanical properties and film-forming ability [19].

Though a core-shell architecture has been claimed in many studies that have reported spherical block copolymer nanoparticles, morphological characterization is usually insufficient [20]. In most cases, the justification for core-shell structure is based only on thermodynamic incompatibility between blocks or their difference of solubility in the continuous phase. There are only few examples where core and shell components have been visually identified and differentiated, mostly through TEM [21,22,23]. Other imaging techniques such as SEM [24] or atomic force microscopy (AFM) [25] have been generally overlooked. Very recently, Gianneschi and coworkers showed the ability of liquid cell TEM to visualize copolymer micelles [26,27] and initiate their formation in situ [28]. The barriers to imaging particle core-shell architecture include small diameters (sometimes well below 100 nm), low differential contrasts, and limited thicknesses of the lyophilic shell. Considering the growing number of studies on particles prepared by PISA [29] and, more generally, the recent developments of polymeric micelles and nanoparticles in experimental medicine and pharmaceutical sciences [30], there is a need to carry out a thorough review of the imaging techniques to substantiate claims of core-shell architecture.

In this study, we started from soft poly(hydroxyethyl acrylate)-*b*-poly(styrene) (PHEA–*b*–PS) particles prepared via dispersion PISA. Relying on a recently published study on PHEA*_x_*–*b*–PS*_y_* nanoparticles [12] (where *x* and *y* refer to the DP of the respective blocks), we chose a copolymer composition located in the spherical aggregate regime for reasons of simplification. Subsequently, model nano-objects with different lengths of PHEA shell-forming blocks were studied by three electron microscopy (EM) techniques: conventional TEM with and without staining, cryo-TEM, and SEM. The objectives of this present study were twofold: to learn which techniques are appropriate for imaging the core-shell architecture of PHEA–*b*–PS particles and, more generally, to define the best practices for the EM analysis of PISA-prepared particles. This is very important because the polymer community is currently making intensive use of EM to study soft block copolymer nanoparticles without always being aware of common errors in the interpretation of EM data like artefacts, sample reorganization induced by drying, and image distortion caused by radiation damage.

## 2. Materials and Methods

### 2.1. Materials

Ruthenium (III) chloride hydrate (Acros Organics; 35–40% Ru), sodium hypochlorite (Acros Organics; 10–15% active chlorine), uranyl acetate (Sigma-Aldrich), and diethyl ether (Sigma-Aldrich; ≥99%) were used as received. PHEA*_x_*–*b*–PS*_y_* copolymer latexes had a solid content of 18.7 wt % and were dispersed in a mixture of methanol/water (95/5 wt %). PHEA_23_–*b*–PS_130_, with the number average molecular weight *M*_n_ = 18.0 × 10^3^ g mol^−1^ and molecular weight dispersity *Đ = M*_w_/*M*_n_ = 1.11, were determined by size exclusion chromatography (SEC) in *N,N*-dimethylformamide. The DP of each block was calculated by ^1^H NMR spectroscopy in D_2_O (PHEA block) and CDCl_3_ (PS block). The z-average value of 79 nm was determined using dynamic light scattering (DLS). For PHEA_85_–*b*–PS_130_, *M*_n_ = 23.1 × 10^3^ g mol^−1^ and *M*_w_/*M*_n_ = 1.43 were determined by SEC in DMF. The z-average hydrodynamic diameter was estimated at 40 nm (DLS data). For each diblock copolymer, SEC traces and DLS size distributions are provided in Appendix A. The synthesis protocol and complete details on characterization methods were described elsewhere [12], and they are briefly summarized in the Appendix A.

### 2.2. Methods

DLS: The particle diameter and size distribution of diblock copolymer nano-objects were determined using a VASCO nanoparticle size analyzer (Cordouan Technologies, Pessac, France) with a 15 mW laser operating at a wavelength of 658 nm. The scattered light was detected at an angle of 135°. Prior to measurements, samples were diluted 20 times.

TEM: The TEM images of the copolymer nanoparticles were obtained with a JEOL ARM-200F instrument working at 200 kV. The images were recorded with Gatan camera (Orius 1000 model). Specimens were prepared on 400 mesh gold grids on which Formvar and carbon support films were applied (Agar scientific, ref. AGS162A4). The number average diameter ($D_{n}=\sum^{\;}D_{\mathrm{TEM}}/n$ where *n* is the number of particles) was determined by using 200 particles. The polydispersity index (PDI) (or dispersity, as recommended by IUPAC) was used to describe the breadth of particle size. The PDI is defined as $D_{w}/D_{n}$, where $D_{w}=\sum^{\;}D_{n}^{4}/\sum^{\;}D_{n}^{3}$. The TEM grid underwent three types of specimen preparations before analysis.
-For dried samples, latex samples were diluted 200 times with a methanol/water mixture (to get a concentration of 0.1 wt %), cast onto the TEM grid, and dried overnight at room temperature.-For positive staining TEM, ruthenium tetroxide (RuO_4_) vapors was produced in situ by reacting 0.5 mL of a 13 wt % aqueous solution of sodium hypochlorite with 150 mg of RuCl_3_∙3H_2_O in a 10-cm diameter petri dish. A TEM grid with the disposed sample (the preparation of the grid according to the same procedure as for dried samples/conventional TEM) was placed close to the reaction mixture, and a second petri dish of similar size was positioned above to form a closed chamber. The grid was left for 10 min. A scheme of the experimental set-up is available in the Appendix A [31].-For negative staining TEM, a rapid flushing method was implemented. The protocol was originally developed by Imai et al. [32] and more recently adapted by Scarff et al. [33]. The idea of the method is to minimize the time the sample has to interact with the support of the grid surface before fixation. The goal is to hinder structural changes in the specimen that could occur upon prolonged absorption time on the carbon film or through capillary action. Before sample application, the TEM grid was faced upon a microscope slide and then irradiated in a glow discharge unit (UV/Ozone ProCleaner Plus) for a minimum of 300 s to render it hydrophilic. As in a typical procedure, 70 µL of uranyl acetate (1 wt % solution in water) was drawn up into the tip of a 200 µL pipette; 10 µL of the air gap were subsequently drawn up, then a 20 µL of deionized water (acting as wash/mixing agent) followed by another air gap of 10 µL, and finally 10 µL of sample solution (1 wt %). The edge of the grid was gripped with a pair of negative pressure tweezers, holding the tweezers so that the grid was angled at approximately 45° facing away from the researcher. The entire content of the pipette tip was ejected across the face of the TEM grid. The excess of stain was removed by touching the torn edge of a piece of filter paper to the edge of the grid. The grid was left to dry over air. Due to difficulties to efficiently adsorb PHEA_23_–*b*–PS_130_-based nanoparticles with the latter protocol, preparation conditions were changed: a 5 µL drop of ethanol was applied onto 400 mesh Cu grids covered with a plain carbon film. After a 1 min interaction, the excess was removed and a 5 µL drop of latex (0.1 wt %) was applied. After 1 min, the excess was removed by touching the torn edge of a piece of filter paper to the edge of the grid, and a 5 µL drop of 2 wt % uranyl acetate aqueous solution was immediately added. After 1 min, the grid was fully dried with a piece of filter paper, and a Tecnai G2 microscope (FEI) operating at 200 kV was used for the imaging of this PHEA_23_–*b*–PS_130_ sample.

Cryo-TEM: Five microliters of the sample (0.1 wt %) were applied onto a 400 mesh Cu grid covered with a lacey carbon film that was freshly glow discharged to render it hydrophilic (Elmo, Cordouan Technologies). The grid was rapidly plunged into a liquid ethane slush by using a homemade freezing machine with a controlled temperature chamber. The grids were then mounted onto a Gatan 626 cryoholder and observed under low dose conditions on a Tecnai G2 microscope (FEI) operating at 200 kV. The images were recorded with a slow scan CCD camera (Eagle 2k2k FEI). The uncertainty of particle diameters were *u*^PHEA85-*b*-PS130^ = 1.30 nm and *u*^PHEA23-*b*-PS130^ = 1.70 nm.

SEM: High resolution SEM images were obtained with a JEOL JSM-7900F scanning electron microscope in gentle beam super high resolution (GBSH) mode. In this mode, an accelerating voltage of 5.5 kV and a specimen bias voltage of −5 kV specimen were applied. The scanning transmission electron microscopy (STEM) images were acquired at 30 kV on the same microscope using the DEBEN GEN5 annular STEM detector. The previously prepared TEM grids were reused for the SEM and STEM observations.

## 3. Results and Discussion

### 3.1. Synthesis of Amphiphilic Diblock Copolymer Nanoparticles PHEA-b-PS

Two types of model amphiphilic diblock copolymer nanoparticles, each containing a constant DP of PS core-forming block (130) and varying DPs of the PHEA stabilizing block (23 and 85), were synthesized with an alcohol-based PISA method described elsewhere [12]. The compounds had a relatively narrow molecular weight dispersity: *Đ = M*_w_/*M_n_* = 1.11 (PHEA_23_–*b*–PS_130_) and 1.43 (PHEA_85_–*b*–PS_130_). In light of the DLS data, the samples were found monodisperse, with z-averages (intensity weighted mean hydrodynamic diameter) of 79 nm (PHEA_23_–*b*–PS_130_) and 40 nm (PHEA_85_–*b*–PS_130_). Using Mie theory and the optical properties of nanoparticles, number-weighted average diameters were also estimated at 39 nm (PHEA_23_–*b*–PS_130_) and 21 nm (PHEA_85_–*b*–PS_130_)—essentially for a comparative purpose with EM data.

Due to the highly hydrophobic nature of PS, the growth of a polar PHEA block in a methanol/water mixture leads to the selective solvation of the chains, inducing phase separation and particle formation. As sketched schematically in Figure 1, the expected core shell architecture was thus made up of a hard lyophobic PS core, a soft lyophilic PHEA shell, and an interfacial region within which the composition varied continuously from one bulk phase to the other. The core-shell architecture of these two PHEA–*b*–PS diblock copolymer nanoparticles of PHEA_85_–*b*–PS_130_ and PHEA_23_–*b*–PS_130_ were systematically characterized using three different EM techniques—conventional TEM (unstained samples, positively stained samples, and negatively stained samples), cryo-TEM, and SEM—emphasizing their own advantages and limitations. A comparative study enabled to assess how the DP of the PHEA stabilizing block affected nanoparticle morphology.

### 3.2. Conventional TEM

#### 3.2.1. Unstained Samples

We began with a TEM analysis of air-dried unstained specimens obtained from PHEA–*b*–PS diblock copolymer nanoparticles. Conventional TEM is indeed the most straightforward and widely applied method for studying the structure of PISA-derived particles, although it can also suffer from non-negligible problems. Its main obstacle is the poor contrast between the different morphological features. Contrast is hindered by the fact that most polymer blocks are amorphous, similar in electron density, and have a limited scattering cross-sections [34]. In addition, the need for specimen drying or high-energy radiation imposed by conventional TEM may create some artefacts by altering the sample’s structure, topography, or chemical composition. This may particularly be the case when block copolymer nanoparticles contain a polymer segment with a sub-ambient glass transition such as PHEA.

Figure 2 shows high magnification TEM images of PHEA_85_–*b*–PS_130_ (Figure 2a) and PHEA_23_–*b*–PS_130_ (Figure 2b) dried nanoparticles obtained without staining preparation. To prove that these micrographs are representative, low magnification images are available in the Appendix A. The two objects commonly exhibit a spherical shape and a narrow particle size dispersity (PDI ≤ 1.05). However, the particles also display a number of differences, particularly in regard to size, morphology, the state of aggregation, and surface structure, that merit more thorough discussion.

Size: In both instances, the number average diameters ($D_{n}$) were well below 100 nm. PHEA_85_-*b*-PS_130_ featured significantly smaller sizes than PHEA_23_–*b*–PS_130_: 19.7 ± 3.0 and 36.4 ± 4.8 nm, respectively. The difference of particle size can be attributed to the length of the lyophilic block that determined particle stability against coalescence. Measurement uncertainty was evaluated for each sample: *u*^PHEA23-PS130^ = 0.5 nm and *u*^PHEA85-PS130^ = 1.4 nm. The values showed that reliable particle sizing is possible. The fact that PHEA_85_–*b*–PS_130_ particles appeared with less details and sharpness accounted for the higher uncertainty value.

Morphology: A close inspection of TEM images revealed differences other than particle diameters. While Figure 2a shows a single particle morphology, Figure 2b shows a vast majority of homogenous spherical particles (accounting for more than 95% of objects) and a few isolated vesicles readily recognizable through their inside space (lumen) appearing brighter (indicated by arrows in Figure 2b). The presence of this second minor population suggested the beginning of an order−order morphological transition induced by a decrease of the PHEA stabilizer DP from 85 to 23. As shown in previous reports [12], sphere-to-vesicle transitions could have resulted from the decrease of the effective volume fraction of the hydrophilic block, that hence increases the packing parameter, *P*, for the copolymer chains. We checked that a new decrease of stabilizing block DP from 23 to 17 resulted in the full transition to vesicle morphology (see the images of PHEA_17_–*b*–PS_136_ particles in Appendix A). This conclusively supported our hypothesis that a PHEA DP of 23 is probably close to the sphere/vesicle phase boundary.

Aggregation state: It is noteworthy that only the latex exhibiting the shortest PHEA block (Figure 2b) showed signs of coalescence. Our interpretation was that there were drying-induced artefacts due to insufficient steric stabilization. Firstly, because much less aggregated particles were visible when decreasing particle concentration (Appendix A). Secondly, DLS data did not show any evidence of aggregates (Appendix A).

Surface structure: Of particular interest is that no domain formation assigned to shell and core regions were visible in TEM images. However, clear differences in the aspect of particle surface could be seen. Only PHEA_85_–*b*–PS_130_ particles boasting the long PHEA stabilizing block showed unsharp particle boundaries, while the PHEA_23_–*b*–PS_130_ particles displayed relatively sharp margins. This result raises two questions. Firstly, were the dried samples representative for the native samples present in dispersion? Secondly, did visible particles boundaries reflect the entire core-shell architecture? Because the lyophilic PHEA phase had a sub-ambient $T_{g}^{\mathrm{PHEA}}$ (−7.3 °C) (see differential scanning calorimetry traces in Appendix A and summary data in Appendix A), the shell was susceptible to significant structural changes upon drying. Our assumption was that PHEA segments could collapse upon drying and form a thin film surrounding the PS core. Under these conditions, the poor mass–thickness contrast of the PHEA phase could have prevented its visualization, and only the PS core (or PS rich domains) may have been imaged. Because PS chains possess a much higher $T_{g}^{\mathrm{PS}}$ (99.0 °C), distortions induced by desolvation and exposure to electrons were minimized. Therefore, in conventional TEM, image contrast from the background originated from the locally increased thickness of the bumpy PS core. While the lyophilic shell cannot be unambiguously imaged, the lack of surface sharpness for PHEA_85_–*b*–PS_130_ particles compared to PHEA_23_–*b*–PS_130_ particles may have translated to a higher excluded volume for the longer PHEA block.

In summary, the TEM analysis of dried particle samples only evaluated the dimension of the core due to collapse of the lyophilic shell upon drying. Hence, the misinterpretation of TEM data could have resulted in errors in the determination of the particle diameter.

#### 3.2.2. Positive Staining

To overcome the aforementioned limitations of conventional TEM, heavy-element-positive staining can be used in order to increase specimen contrast and possibly resolve the PHEA shell unseen with unstained samples. Only in a few isolated studies have staining methods been used to preferentially visualize one phase of core-shell copolymer nanoparticles [35,36,37,38,39]. In the case of positive stain EM, the higher contrast is generated primarily by the differential electron scattering due to more electron-dense stained regions inside the particles and a less electron-dense unstained surrounding environment.

Figure 3 shows the same PHEA_85_–*b*–PS_130_ (Figure 3a) and PHEA_23_–*b*–PS_130_ (Figure 3b) samples after RuO_4_-positive staining. RuO_4_ is a strong scattering agent that reacts with polymer regions inside the particle, thus improving the contrast [40]. Because of its high oxidative properties, RuO_4_ is presumed to stain both PHEA and PS domains by reacting, respectively, with hydroxyl groups and aromatic unsaturations [31,41,42]. Four different structural features can be highlighted:

Contrast: Regardless of the sample, the action of the stain considerably improved the image contrast against the support film. Image contrast against a grid background can be further enhanced after latex dialysis. In this case, the sample was devoid of soluble copolymer chains (Appendix A). As noted previously, PHEA_23_–*b*–PS_130_ exhibited a pronounced tendency to form pools of clustered particles, while PHEA_85_–*b*–PS_130_ particles were well separated.

Size: Interestingly, an increase of the average diameter $D_{n-{\mathrm{RuO}}_{4}}^{}$ was also observed compared to unstained analogues: 50% for PHEA_85_–*b*–PS_130_ (31.1 ± 3.8 nm) and 15% for PHEA_23_–*b*–PS_130_ (41.8 ± 7.2 nm). This increase was consistent with our hypothesis that the superficial PHEA-rich region (shell) might be undetected with the unstained specimen and required staining to be revealed. As further evidence, we noted that the size increase for PHEA_85_–*b*–PS_130_ particles was much larger than for PHEA_23_–*b*–PS_130_, which was consistent with the longer PHEA block and more extended lyophilic shell. It is worth noting that the diameter of the positively stained specimen significantly departed from the number-weighted mean diameters estimated by the DLS measurements: 21 nm (PHEA_85_–*b*–PS_130_) and 39 nm (PHEA_23_–*b*–PS_130_.). It has been well-reported that a direct comparison between the TEM of dried particles and DLS data involving solvated particles is not always relevant [43].

Halo: As shown in Figure 3a, the positively stained PHEA_85_–*b*–PS_130_ showed a darker halo contouring the spherical particles. It is therefore tempting to interpret this feature as a sign of a core-shell architecture. However, the second sample with the short PHEA did not display the similar darker annular contrast, and a control experiment with pure PS nanoparticles (not dialyzed) of similar size subjected to the same staining procedure again showed the same artefact (Appendix A). The inner ring of darker contrast has been already described in the literature [44] and can be related to two issues: a higher stain concentration at particle surface that makes the particles boundaries more scattering [35] or the result of drying with the preferential adhesion of organic residue on particle surface against the TEM support [45]. In our case, the purification of particles by dialysis did not remove the dark ring (Appendix A), suggesting the minor role played by stain surface accumulation.

Morphology: Since RuO_4_ reacts with both polymer segments, it is not possible to achieve a non-ambiguous differentiation of the particle core and shell. In order to preferentially stain the shell, OsO_4_ was used because of its inefficiency to oxidize the PS phase. Unfortunately, the images (Appendix A) were similar to those of the unstained samples (Figure 2), indicating that OsO_4_ staining was ineffective and could not be selectively incorporated in one individual component.

In summary, an RuO_4_ stain cannot reveal the multiphase structure of composite PHEA–*b*–PS particles. However, this positive staining is able to shape boundaries of polymer particles, which is of high interest to get a complete picture of the particle. However, there are justified doubts about positive staining’s ability to preserve the native structure of the specimen, as well as a trend to overestimate particle diameter.

#### 3.2.3. Negative Staining

Another complementary way to analyze soft nanomaterials is negative staining TEM, which has been very popular for biological structures but much less employed for polymer colloids [35,37]. In negative staining, the stain does not penetrate the particles; rather it adsorbs to the surface and is dissolved in the continuous phase. After solvent evaporation, the particles are enveloped by an amorphous matrix of an electron-dense stain compound, with the advantages of increasing the contrast; the image is therefore a kind of finger-print [46,47]. When it is properly performed, negative staining with uranyl acetate (UAc) can also preserve the sample’s integrity, morphology, and size.

Figure 4 represents typical TEM images of two negatively stained samples derived from PHEA_85_–*b*–PS_130_ (Figure 4a) and PHEA_23_–*b*–PS_130_ (Figure 4b) using UAc as the negative stain. A few remarks can be made as dominant characteristics of negatively stained images.

Contrast: As an indication that negative stain was successful, a contrast was produced between the background and the particles. The particles appeared as light areas because of their low electron scattering power relative to the dense surrounding stain, which scattered the electrons more and appeared darker. However, the negative stain was unable to penetrate the object, so internal structural details could not be deduced. Clearly, PHEA_23_–*b*–PS_130_ was more amenable to visualization by this method than PHEA_85_–*b*–PS_130_. In this case, the copolymer assemblies were more fragile because of a poorer steric stabilization caused by a shorter PHEA stabilizing block. As a result, the particles were more likely to collapse or disassemble upon adsorption, staining, or drying on the grid.

Size and shell thickness: The number average diameters $D_{n-\mathrm{UAc}}^{}$ = 29.7 ± 4.2 nm (PHEA_85_–*b*–PS_130_) and 41.3 ± 5.1 nm (PHEA_23_–*b*–PS_130_) were substantially larger than those of the unstained samples. This suggested that a negative coating at particle surface may have been efficient in revealing the particle boundaries, which is a requirement for precise measurement of particle size. Slightly smaller values were found compared to positively stained samples for PHEA_85_–*b*–PS_130_ (−5.1 nm) but not for PHEA_23_–*b*–PS_130_. This could indicate that positive staining may have over-estimated the dimensions of the nano-objects but only when the shell was extended. By comparing the negatively stained images ($D_{n-\mathrm{UAc}}^{}$) where particles were presented in the most faithful way and the unstained image $\left(D_{n}^{}\right)$ was only representative of the particle core, the thickness *δ* of the shell could be tentatively estimated (*δ* = $(D_{n-\mathrm{UAc}}^{}\;$ − $D_{n}^{})/2$). For PHEA_85_–*b*–PS_130_, *δ* ~ 5.0 ± 1.8 nm, while the δ of PHEA_23_–*b*–PS_130_ was in the error range and could not be accessed with precision due to the limited extent of the PHEA domain. PHEA_85_-*b*-PS_130_ was composed of 85 repeat units for the lyophilic backbone segment, which translated to a maximum end-to-end distance of 15 nm for a fully stretched polymer chain. This value was thus much larger than that of the new region revealed by negative staining. Two reasons are suggested to explain this gap: Firstly, PHEA chains were not likely to be fully stretched because this conformation was entropically unfavorable, and secondly, this soft shell was susceptible to contraction by drying despite negative staining.

In summary, negative staining may be more efficient in preserving the pristine state of copolymer nanoparticles and may shape the particular boundaries in a more faithful manner than positive staining. A comparison with an unstained image can give an indirect access to shell thickness when spatial resolution is sufficient. However, no all specimens are amenable to visualization by negative staining, and core-shell phase differentiation remain inaccessible.

### 3.3. Cryo-TEM

Previous analyses have shown that PHEA–*b*–PS particles are problematic specimens for conventional TEM, in particular due their proneness to dehydration in the vacuum of the electron microscope. In what follows, the two same samples were analyzed by cryo-TEM, a useful technique to visualize particle in their dispersion state by rapid vitrification in a thin layer of solvent. Though cryo-TEM requires more complex specimen preparation, it is the only technique that is able to image nanoparticles in their native form.

Figure 5 represents two typical cryo-TEM images of PHEA_85_–*b*–PS_130_ (Figure 5a) and PHEA_23_–*b*–PS_130_ (Figure 5b) nanoparticles at low magnification.

Size: We noticed that the average diameter of the PHEA_23_–*b*–PS_130_ particles (37.0 ± 5.5 nm) was relatively similar regardless of the preparation method: 36.4 nm with drying and 41.8 and 41.3 nm with positive and negative staining, respectively. In this case, the shell was thin (<2 nm), resulting in minimal size deviations between techniques that could resolve PHEA region (negative/positive staining TEM) and the others that could not. Conversely, PHEA_85_–*b*–PS_130_ presented a more extended shell, and its apparent diameter was consistently more dependent on the EM preparation method. Interestingly, the particle size calculated from the cryo-TEM image (21.6 ± 3.2 nm) was relatively similar to that of unstained dried sample (19.7 nm) and, was therefore substantially lower than the diameters derived from positively stained TEM (31.1 nm) and negatively stained TEM (29.7 nm). This suggests that analysis of undried specimen by cryo-TEM also only evaluated the dimension of the core. This time, the explanation was not the collapse of the hydrophilic shell but the difference of the conformation between lyophilic and lyophobic chains. The PHEA chains were likely to be partially stretched, whereas the PS core was more closely packed due to its hydrophobic nature. The difference of chain density between core and shell chains accounted for the mass–thickness contrast of PHEA significantly smaller than that of PS, leading to a poorer capacity to scatter electrons.

Morphology: A more in-depth examination of the PHEA_85_–*b*–PS_130_ specimen obtained at high magnification (see the inset of Figure 5a) revealed dark spots inside the particles. These features were not artefacts and could be rationalized on the basis of extended PHEA chains pointing in a direction parallel to the electron beam. Conversely, chains that were not parallel to the beam had a much lower scattering efficiency and could not be imaged. Due to fast vitrification, the PHEA molecules usually adopt random orientations in the amorphous ice layer; consequently, the PHEA chains at the edge of the particle were unlikely to adopt this particular orthogonal orientation, making the visualization of lyophilic shell impossible. In contrast, PHEA_23_–*b*–PS_130_ did not display a similar patched structure since the short PHEA chains did not have a sufficient scattering efficiency. In the same sample, it was noteworthy that some isolated vesicles could be detected in the inset of Figure 5b, supporting that their presence in conventional TEM was not an artefact caused by drying or concentration effects. In addition, no sign of coalescence was visible (as noted previously in TEM images), thus confirming the previous assumption of drying-induced aggregation (vide supra).

Surface structure: Figure 5a shows that PHEA_85_–*b*–PS_130_ particles presented a fuzzy surface that resembled an envelope, as opposed to the sharper surface of PHEA_23_–*b*–PS_130_ (Figure 5b) and PS particles (Figure 5c) serving as benchmark. Used as benchmark latex, the particles appeared homogeneous with a smooth surface. The fog surface, together with the irregular shape of the copolymer nanoparticles, could be assigned to a brush structure with density fluctuation as opposed to a dense core. This feature was reminiscent of a core-shell architecture, although no indisputable proof was provided because no morphology domains could be imaged within the particle.

In summary, cryo-TEM cannot resolve the nanoparticle shell because the lyophilic PHEA in stretch conformation forms regions with a low chain density and therefore poor electron capacity. When a shell-forming block is long enough, its presence is manifested by an unsharp envelope and dark spots inside the PS core suggestive of chains perpendicular to the plane.

### 3.4. SEM

SEM has distinctive advantages compared to TEM and cryo-TEM; in particular, finer surface structure images can be obtained upon operating at lower accelerating voltages. Under these conditions, the penetration and diffusion area of incident electrons is shallow. Surface structures are thus gained because the number of secondary electrons emitted from the surface is maximized compared to backscattered electrons generated from within the specimen [48]. SEM can be employed to analyze nanoparticle size, shape, and surface structure.

Figure 6 displays SEM images of PHEA_85_–*b*–PS_130_ (Figure 6a) and PHEA_23_–*b*–PS_130_ (Figure 6b) nanoparticles. To ease interpretation, a control analysis with PS nanoparticles was also carried out (Figure 6c). To prove that these micrographs are representative, low magnification images are also available in the Appendix A. In all instances, an accelerating voltage of 5.5 kV was used during image acquisition in order to emphasize surface structures. Thanks to a −5 kV specimen bias voltage, the electrons eventually struck the sample with a landing voltage of only 500 V, thus protecting it from damage.

Particle size: The number average particle diameters of PHEA_85_–*b*–PS_130_ (22.9 ± 2.9 nm) and PHEA_23_–*b*–PS_130_ (34.8 ± 5.3 nm), as well as size dispersities (PDI ≤ 1.04), were comparable to those of TEM (without staining) and cryo-TEM. This result suggested that SEM also failed to resolve the PHEA shell of the PHEA_85_–*b*–PS_130_ specimen. However, particle core size and shape could be determined with precision. Like previous TEM images, the PHEA_23_–*b*–PS_130_ latex exhibiting the shortest PHEA block showed signs of coalescence attributed to drying.

Surface structure: Using the PS latex as a benchmark, a thin bright corona attributed to edge effects are visualized in Figure 6c [49]. It is well-established that particle edges (see schematic explanation in Appendix A) allows for greater electron beam penetration into surface region. This results in the generation/escape of a larger number of secondary electrons, giving rise to a typical bright particle surface. By comparison, PHEA_23_–*b*–PS_130_ (Figure 6b) displayed a much thicker bright corona due to the emission of more secondary electrons. PHEA_85_–*b*–PS_130_ particles (Figure 6c) appeared even brighter because the surface may have had a higher secondary electron emitting capacity. The more pronounced brightening of the copolymer particles compared to the PS particles could be reconciled with the presence of protrusions onto the particle surface. The longer the PHEA chains, the greater the edge effect. Another non-direct indication of core-shell morphology relies on extremely blurred interphase, particularly for PHEA_85_–*b*–PS_130_-based nanoparticles.

Scanning transmission electron microscopy (STEM): The same tandem of nanoparticles (PHEA_85_–*b*–PS_130_ and PHEA_23_–*b*–PS_130_) was also analyzed using an STEM detection mode. This transmission working mode is known to provide a better spatial resolution than SEM because of a stronger mass–thickness contrast. Accordingly, measurement uncertainties were of the same order of magnitude or even weaker than for TEM data: $u^{\mathrm{PHEA}85-b-\mathrm{PS}130}$ = 0.70 nm and $u^{\mathrm{PHEA}23-b-\mathrm{PS}130}$ = 0.85 nm. In addition, the accelerating voltage was much lower than in conventional TEM (30 kV versus 200 kV), implying fewer risks of damaging the sample, especially the PHEA layer. The results gathered in Table 1 show a strong resemblance between STEM and TEM size in regard to average particle diameter and measurement uncertainty. Again, staining revealed the entire particle volume without contrasting shells and cores, while non-stained samples showed only the PS core. All STEM images are available in Appendix A.

In summary, the edge effect in SEM might have been evidence of the presence of protruding PHEA chains at the particle surface (although more evidence is needed). Nevertheless, the PHEA shell could not be resolved in a way to allow for the precise determination of particle size and shape. Therefore, the apparent diameter was largely the particle core. Interestingly, the STEM mode could be considered a viable alternative to TEM to image the block copolymer nanoparticles.

## 4. Conclusions

Amphiphilic block polymer nanoparticles display a core-shell architecture, but the precise imaging of their multiphase structure has long been elusive and problematic. Herein, we reported the morphological characterization of two PHEA*_x_*–*b*–PS_130_ copolymer nanoparticles bearing short and long PHEA stabilizing blocks (*x* = 23 and 85) using TEM, cryo-TEM, and SEM. The shell extent of PHEA_23_–*b*–PS_130_ was too short to be resolved, regardless of the method. This explains why the particle size given by the three EM techniques were in the same average order. By contrast, the PHEA_85_–*b*–PS_130_ particles boasting the longest PHEA stabilizing block showed significant differences depending on the technique. The TEM analysis of air-dried unstained sample could only image the particle core, and a blurred interface provided the hint of the shell existence. Staining is thus essential for shaping particle boundaries, and negative staining using uranyl acetate seems more amenable than positive staining (RuO_4_) to fix particles in their pristine form and then obtain a precise measurement of particle size. However, here, no staining method was able to reveal the multiphase structure. Cryo-TEM ensured that the specimens could be studied in their native form, but the solvophilic PHEA in stretch conformation exhibited too poor an electron capacity to allow for visualization, and only protruding chains appeared as dark spots from the particle interior. Like TEM and cryo-TEM, SEM could only evaluate the PS core. However, fine surface structures reminiscent of PHEA shell could be imaged through the “edge effect,” leading to highly bright particles with an unsharp interface. In conclusion, no technique allowed for a satisfactory imaging of core and shell regions, but their combination gave access to particle size, shell thickness, and specific morphological features suggestive of shell existence. We believe that these findings will open new avenue in the rational use of microscopy techniques for the analysis of soft self-assembled polymer nanostructures.

## Figures and Tables

**Figure 1 polymers-12-01656-f001:**
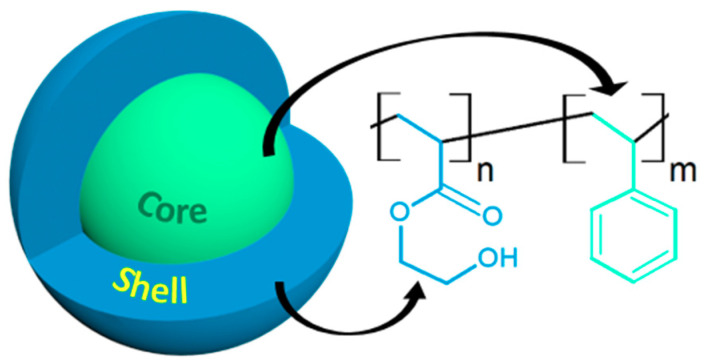
Schematic representation of poly(hydroxyethyl acrylate)–*b*–poly(styrene) (PHEA–*b*–PS) core-shell nanoparticles.

**Figure 2 polymers-12-01656-f002:**
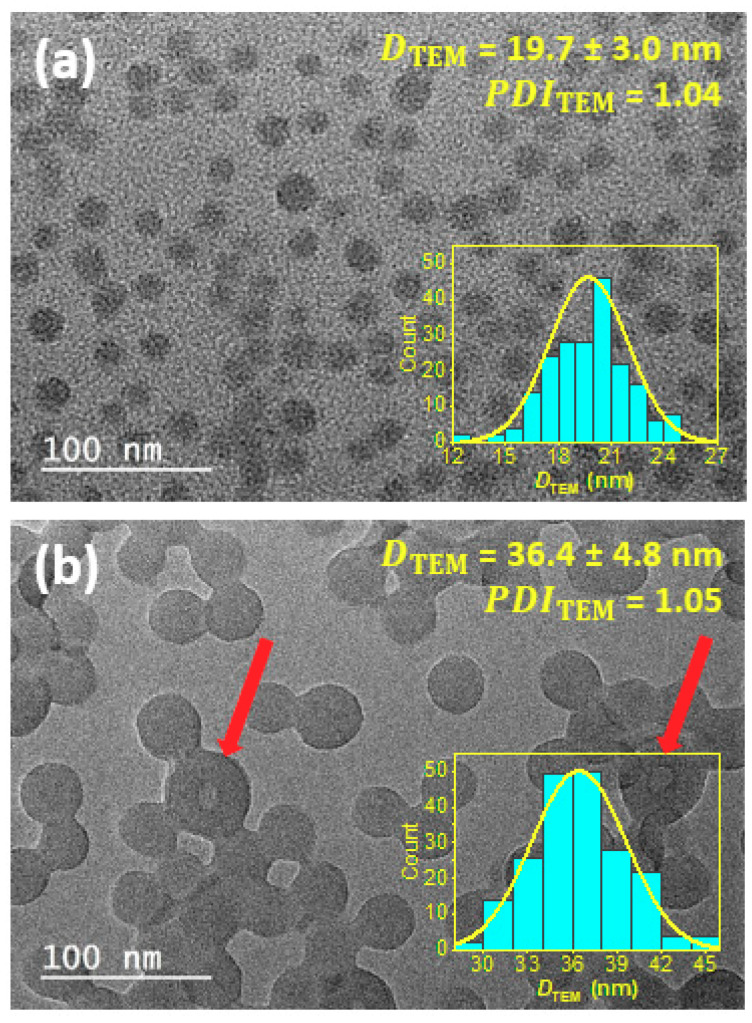
Conventional TEM images of (**a**) PHEA_85_–*b*–PS_130_ and (**b**) PHEA_23_–*b*–PS_130_ block copolymer nanoparticles. The inset shows the size distribution obtained from TEM measurements. The red arrows in (**b)** highlight vesicular structures.

**Figure 3 polymers-12-01656-f003:**
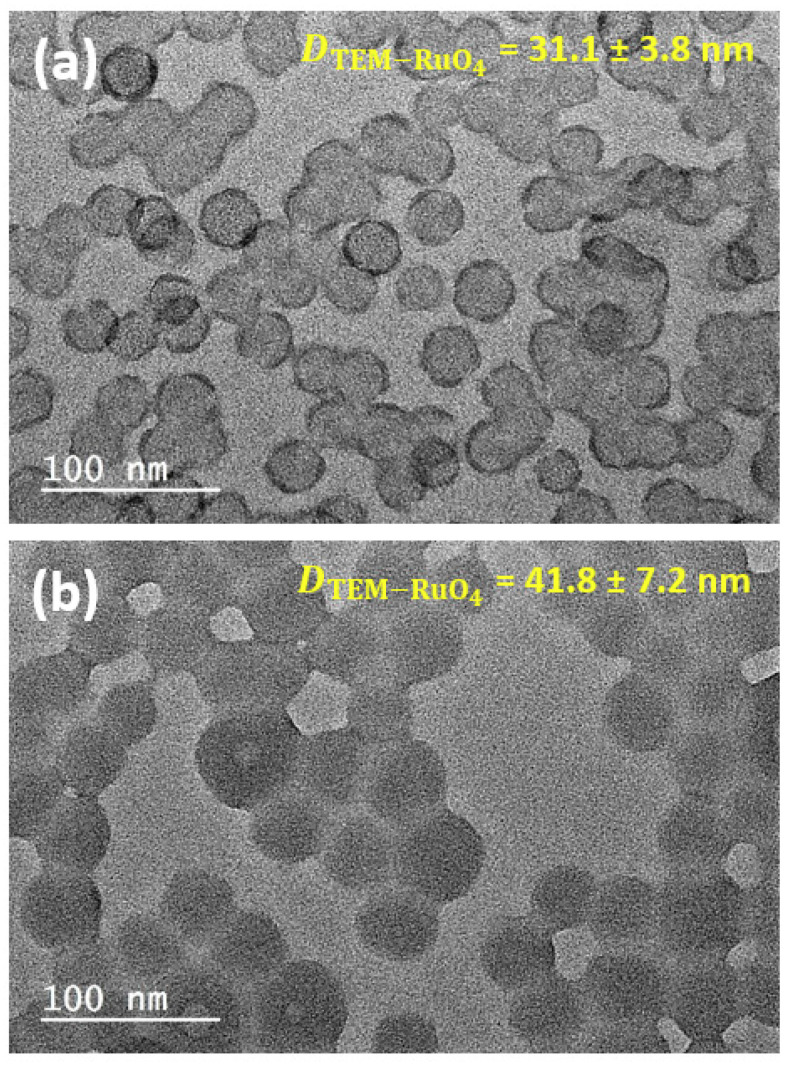
Positive staining electron microscopy (EM) images of (**a**) PHEA_85_–*b*–PS_130_ and (**b**) PHEA_23_–*b*–PS_130._

**Figure 4 polymers-12-01656-f004:**
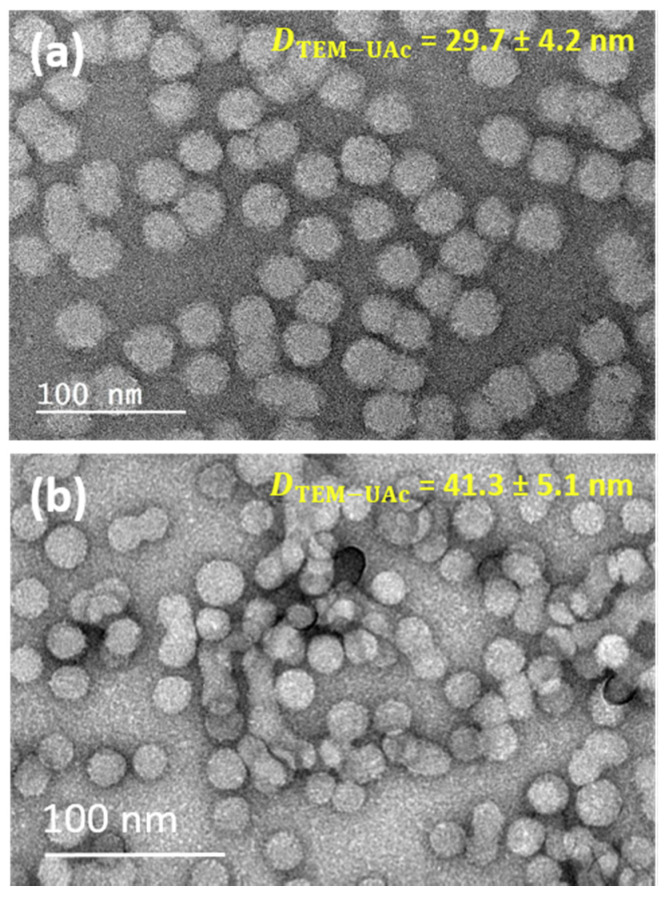
Negative stain electron microscopy of (**a**) PHEA_85_–*b*–PS_130_ and (**b**) PHEA_23_–*b*–PS_130._

**Figure 5 polymers-12-01656-f005:**
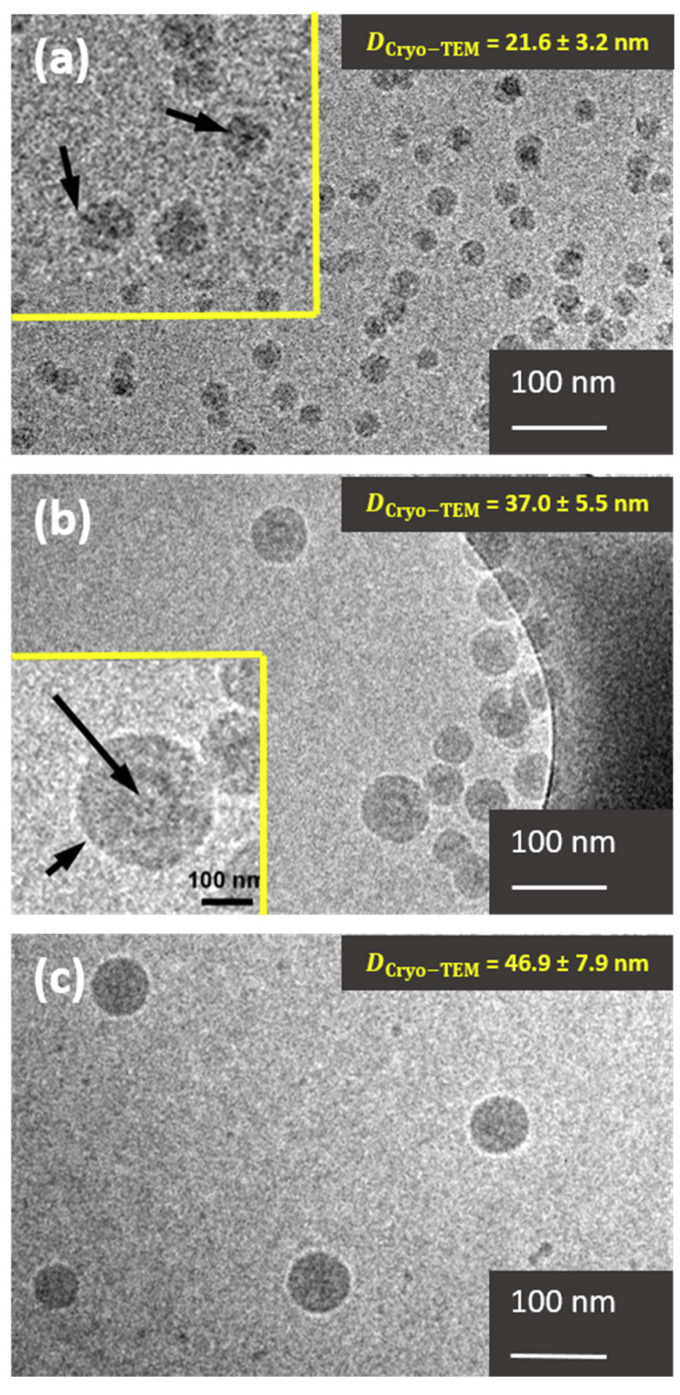
(**a**) Cryo-TEM image of PHEA_85_–*b*–PS_130_ particles. Particle surface displays some dark spots, better seen in the inset. Indicated by arrows, these spots correspond to PHEA chains pointing parallel to the electron beam trough the continuous phase. (**b**) Cryo-TEM image of the PHEA_23_–*b*–PS_130_ particles. Two particle morphologies can be distinguished: spherical particles and vesicles. The inset shows the vesicle. The darker rim corresponds to the PHEA chains (short arrow), then the light grey the PS, and finally, again, a darker thin rim for the PHEA chains in contact with the inner space (long arrow). (**c**) Pure PS nanoparticles.

**Figure 6 polymers-12-01656-f006:**
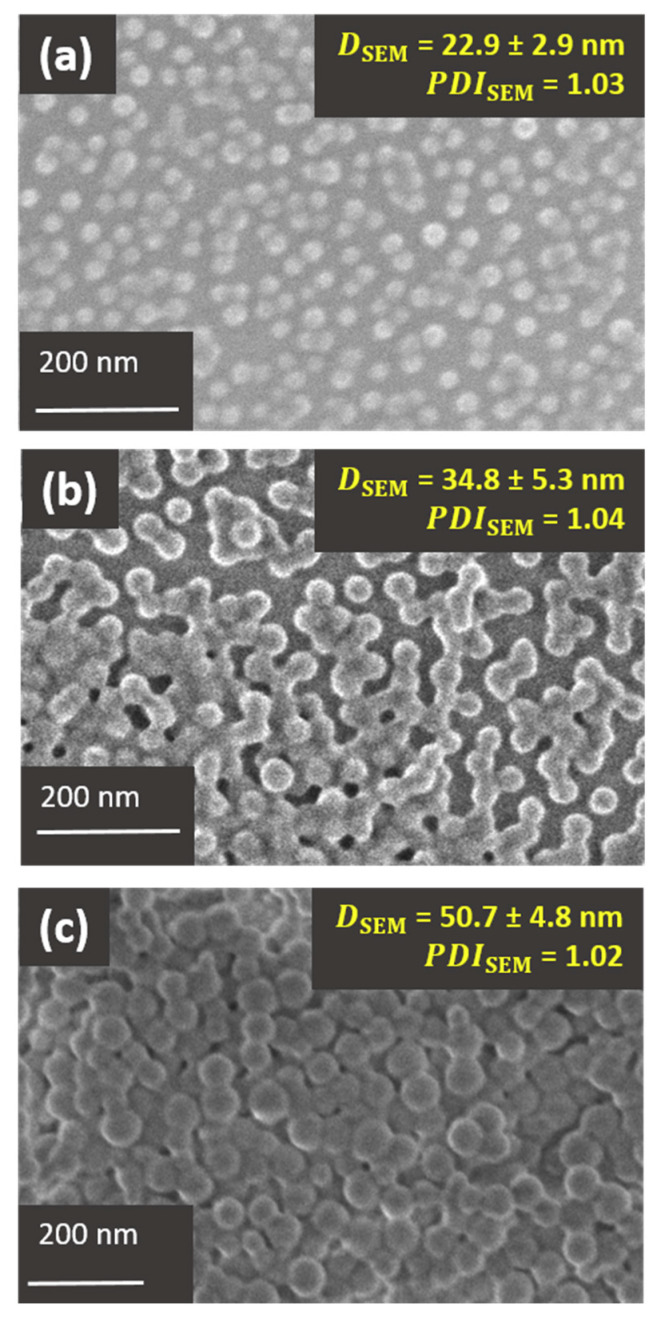
SEM images of (**a**) PHEA_85_–*b*–PS_130_, (**b**) PHEA_23_–*b*–PS_130_ diblock copolymer nanoparticles, and (**c**) PS latex used as model latex. All images were obtained at a landing voltage of 500 V.

**Table 1 polymers-12-01656-t001:** Comparison between particle size data determined by TEM and SEM.

Microscopic Method	PHEA_85_–*b*–PS_130_	PHEA_23_–*b*–PS_130_
	*D*_p_, nm	
TEM	19.7 ± 3.0	36.4 ± 4.8
TEM with RuO_4_-positive staining	31.1 ± 3.8	41.8 ± 7.2
STEM	20.2 ± 1.7	34.8 ± 2.6
STEM with RuO_4_-positive staining	28.1 ± 2.2	42.6 ± 4.0

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
