# Peer review of "Characterizing the Core-Shell Architecture of Block Copolymer Nanoparticles with Electron Microscopy: A Multi-Technique Approach"

_polymers, 2020, doi:10.3390/polym12081656_

Round 1
Reviewer 1 Report
This manuscript compared the observation results of PHEA-b-PSt self-assembled nanoparticles through different EM techniques and proposed one combined route to characterize the morphology of core-shell particles, which is very important and helpful. After minor revision, it is suggested to be accepted for publication.
- “usually missing” should be changed to “insufficient” or other words.
- Materials: it is so strange to determine separately the block lengths of PHEA and PSt. Probably, the block length of PHEA was firstly determined, then the molar ratio of HEA to St was determined in one common solvent. “z-average” should be “z-average hydrodynamic diameter”.
- Line 129-131: why was air gap of 10 μL drawn up?
- As for Figure 1 and the figure in Graphic Abstract: the interfacial phase should be hidden since this manuscript does not deal with it. The title of Figure 1 has an improper expression, PHEA-b-PSt is one block copolymer, which is not a mixture.
- The abbreviates such as STEM (Line 152) should have their full name at the first appearance.
- Line 261: the latex without soluble block copolymer is the exact sample for EM observation.
- Line 277: as for “pure PSt nanoparticles”, was there any purification such as the removal of surfactant?
- The title of Figure 5: the explanation in it should be removed into the manuscript mainbody.
- Line 362: “largely evaluates the dimension of the core” is difficulty to follow.
- Line 419: according to the figure title, “image 6c” is assigned to PSt nanoparticles. Please check it and re-write the related content.
- Type errors: “made up” (Line 15 and 16) → “made up of”; “and shape the entire particle volume” (Line 23) → “, shape and the entire particle volume”; “DP PS core-forming block” (Line 158) → “DP of PS core-forming block”; “account” (Line 207) → “accounts”; “a most faithful way possible particle” (Line 321) → “the most faithful way”.

Author Response
This manuscript compared the observation results of PHEA-b-PSt self-assembled nanoparticles through different EM techniques and proposed one combined route to characterize the morphology of core-shell particles, which is very important and helpful. After minor revision, it is suggested to be accepted for publication.
- “usually missing” should be changed to “insufficient” or other words.
Line 63 page 2. “detailed morphological characterization is usually insufficient” was changed to “morphological characterization is usually insufficient”.
- Materials: it is so strange to determine separately the block lengths of PHEA and PSt. Probably, the block length of PHEA was firstly determined, then the molar ratio of HEA to St was determined in one common solvent. “z-average” should be “z-average hydrodynamic diameter”.
Line 95 page 3. PHEA block length was first determined 1H NMR in D2O, then the length of the additional PSt block obtained by chain extension was obtained by 1H NMR in CDCl3.
- Line 129-131: why was air gap of 10 μL drawn up?
Line 122 page 3. A number of negative staining protocols have been tested for this study, and we’ve found that flushing method was the most suitable to our block copolymer system. The basic idea is to create three separate compartments inside the micropipette containing 1/ the diluted latex ; 2/ water as washing solution ; and 3/ negative stain solution (uranyl acetate in our case). The three solutions are separated by two air gaps of 10 µL. By pushing down the plunger of the micropipette, the three samples are successively dispensed and deposited on the TEM grid. Therefore, the air gap ensures that there is no mixing of the three samples.
- As for Figure 1 and the figure in Graphic Abstract: the interfacial phase should be hidden since this manuscript does not deal with it. The title of Figure 1 has an improper expression, PHEA-b-PSt is one block copolymer, which is not a mixture.
Line 168 page 4. The schematic representation of block copolymer nanoparticles in Figure 1 and Graphical abstract have been changed to remove the interfacial region.
- The abbreviates such as STEM (Line 152) should have their full name at the first appearance.
Line 152 page 4. Abbreviation STEM was defined after the first occurrence.
- Line 261: the latex without soluble block copolymer is the exact sample for EM observation.
Line 260 page 7. This part has been rephrased to emphasize that the dialyzed sample does not contain any free block copolymer chains (unimers).
- Line 277: as for “pure PSt nanoparticles”, was there any purification such as the removal of surfactant?
Line 260 page 7. This sample was not dialyzed. Therefore, it is supposed to contain some free ionic surfactant (sodium dodecyl sulfate in this case). This information was added to the revised manuscript.
- The title of Figure 5: the explanation in it should be removed into the manuscript mainbody
The figure caption is probably too long, and a part of the explanations were transferred to the manuscript mainbody (see lines 369 and 382 page 11). However, the figure being relatively complex, it is useful to keep some details in the caption to ease the understanding. Non-essential information was either removed or moved.
- Line 362: “largely evaluates the dimension of the core” is difficulty to follow.
Line 360 page 10. “largely evaluates the dimension of the core” was changed to ““only evaluates the dimension of the core”. The change was also made at two other occurrences: Line 240 page 6 and Line 458 page 13
- Line 419: according to the figure title, “image 6c” is assigned to PSt nanoparticles. Please check it and re-write the related content.
We confirm that Figure 6c is a SEM micrograph of PSt nanoparticles
- Type errors: “made up” (Line 15 and 16) → “made up of”; “and shape the entire particle volume” (Line 23) → “, shape and the entire particle volume”; “DP PS core-forming block” (Line 158) → “DP of PS core-forming block”; “account” (Line 207) → “accounts”; “a most faithful way possible particle” (Line 321) → “the most faithful way”.
All the typos were corrected. Thank you for this.
Reviewer 2 Report
Comments on “Characterizing the Core-shell Architecture of Amphiphilic Block Copolymer Spherical Nanoparticles: A Multi-Technique Approach” by Vitalii Tkachenko et al.
In this work, the authors have adopted different experimental techniques to characterize the core-shell architecture of the amphiphilic block copolymer spherical nanoparticles, formed by poly(hydroxyethyl acrylate)-b-poly(styrene) through polymerization-induced self-assembly. Both the transmission electron microscopy (TEM) with or without staining, cryo-TEM and scanning electron microscopy (SEM) are used for these characterization studies. The advantages and limitations of these methods are discussed in detail, such as precise shaping of particle boundaries, resolving the particle shell, differentiating particle core and shell, and the effect of sample drying and staining. Overall, this work is well-written and scientific sound. It also provides a solid basis for polymer chemists to develop their own specimen preparations for these different characterization methods. In addition, this work might be further strengthen by discussing the most recent advanced technique, such as liquid cell transmission electron microscopy, which is emerging to characterize these polymeric systems in their native environment.
[1] Touve, Mollie A., C. Adrian Figg, Daniel B. Wright, Chiwoo Park, Joshua Cantlon, Brent S. Sumerlin, and Nathan C. Gianneschi. "Polymerization-induced self-assembly of micelles observed by liquid cell transmission electron microscopy." ACS central science 4, no. 5 (2018): 543-547.
[2] Parent, Lucas R., Evangelos Bakalis, Maria Proetto, Yiwen Li, Chiwoo Park, Francesco Zerbetto, and Nathan C. Gianneschi. "Tackling the challenges of dynamic experiments using liquid-cell transmission electron microscopy." Accounts of chemical research 51, no. 1 (2018): 3-11.
[3] Parent, Lucas R., Evangelos Bakalis, Abelardo Ramírez-Hernández, Jacquelin K. Kammeyer, Chiwoo Park, Juan De Pablo, Francesco Zerbetto, Joseph P. Patterson, and Nathan C. Gianneschi. "Directly observing micelle fusion and growth in solution by liquid-cell transmission electron microscopy." Journal of the American Chemical Society 139, no. 47 (2017): 17140-17151.
Author Response
In this work, the authors have adopted different experimental techniques to characterize the core-shell architecture of the amphiphilic block copolymer spherical nanoparticles, formed by poly(hydroxyethyl acrylate)-b-poly(styrene) through polymerization-induced self-assembly. Both the transmission electron microscopy (TEM) with or without staining, cryo-TEM and scanning electron microscopy (SEM) are used for these characterization studies. The advantages and limitations of these methods are discussed in detail, such as precise shaping of particle boundaries, resolving the particle shell, differentiating particle core and shell, and the effect of sample drying and staining. Overall, this work is well-written and scientific sound. It also provides a solid basis for polymer chemists to develop their own specimen preparations for these different characterization methods. In addition, this work might be further strengthen by discussing the most recent advanced technique, such as liquid cell transmission electron microscopy, which is emerging to characterize these polymeric systems in their native environment.
[1] Touve, Mollie A., C. Adrian Figg, Daniel B. Wright, Chiwoo Park, Joshua Cantlon, Brent S. Sumerlin, and Nathan C. Gianneschi. "Polymerization-induced self-assembly of micelles observed by liquid cell transmission electron microscopy." ACS central science 4, no. 5 (2018): 543-547.
[2] Parent, Lucas R., Evangelos Bakalis, Maria Proetto, Yiwen Li, Chiwoo Park, Francesco Zerbetto, and Nathan C. Gianneschi. "Tackling the challenges of dynamic experiments using liquid-cell transmission electron microscopy." Accounts of chemical research 51, no. 1 (2018): 3-11.
[3] Parent, Lucas R., Evangelos Bakalis, Abelardo Ramírez-Hernández, Jacquelin K. Kammeyer, Chiwoo Park, Juan De Pablo, Francesco Zerbetto, Joseph P. Patterson, and Nathan C. Gianneschi. "Directly observing micelle fusion and growth in solution by liquid-cell transmission electron microscopy." Journal of the American Chemical Society 139, no. 47 (2017): 17140-17151.
We thank the reviewer. It is an excellent point that we have missed. The introduction was revised to include a reference to liquid cell transmission electron microscopy (LCTEM) technique including relevant references. “Very recently, Gianneschi and coworkers showed the interest of liquid cell TEM micelle formation to visualize micelle and initiate in situ their formation.”
Reviewer 3 Report
Dear authors,
Your article on characterizing core-shell morphology of nanoparticles using various EM techniques is very insightful and well written.
After reviewing the manuscript, I have only few comments to improve the way it is presented to the readers.
Line 248, 285: use another adjective instead of decorate
Line 300: First use of UAc is in this line and not 304. Therefore, use the full form with abbreviation in line 300.
Line 318 still contains annotations from previous review that weren't removed before submission.
As a general comment, nanoscale-IR spectroscopy and imaging using an instrument like nanoIR3 from Anasys/Bruker can be further utilized to characterize the nanoparticles to gain complimentary chemical compositional information/confirmation.
Author Response
Your article on characterizing core-shell morphology of nanoparticles using various EM techniques is very insightful and well written.After reviewing the manuscript, I have only few comments to improve the way it is presented to the readers.
- Line 248, 285: use another adjective instead of decorate
Lines 248 page 7 and 285 page 8. The word “decorate” was replaced by “visualize” and “stain”, respectively.
- Line 300: First use of UAc is in this line and not 304. Therefore, use the full form with abbreviation in line 300.
Line 300 page 8. Thank you for this. The abbreviation of uranyl acetate (UAc) was defined after its first occurrence line 300 instead of line 304.
- Line 318 still contains annotations from previous review that weren't removed before submission.
All annotations were removed. Sorry for this inconvenience.
- As a general comment, nanoscale-IR spectroscopy and imaging using an instrument like nanoIR3 from Anasys/Bruker can be further utilized to characterize the nanoparticles to gain complimentary chemical compositional information/confirmation.
The common theme of this paper is electron microscopy. However, we entirely agree that nanoscale IR spectroscopy (not accessible in our institute) could be a valuable tool to image the two polymer phases.